# A Data-Mining Approach to Identify NF-kB-Responsive microRNAs in Tissues Involved in Inflammatory Processes: Potential Relevance in Age-Related Diseases

**DOI:** 10.3390/ijms24065123

**Published:** 2023-03-07

**Authors:** Luigina Micolucci, Giulia Matacchione, Maria Cristina Albertini, Massimo Marra, Deborah Ramini, Angelica Giuliani, Jacopo Sabbatinelli, Antonio Domenico Procopio, Fabiola Olivieri, Annalisa Marsico, Vladia Monsurrò

**Affiliations:** 1Department of Clinical and Molecular Sciences, Università Politecnica Delle Marche, Via Tronto 10/A, 60126 Ancona, Italy; 2Department of Biomolecular Sciences, University of Urbino Carlo Bo, 61029 Urbino, Italy; 3Clinic of Laboratory and Precision Medicine, IRCCS INRCA, 60121 Ancona, Italy; 4Laboratory Medicine Unit, Azienda Ospedaliera Universitaria “Ospedali Riuniti”, Via Conca 71, 60126 Ancona, Italy; 5Institute for Computational Biology, Helmholtz Centre, 85764 Munich, Germany; 6Max Planck Institute for Molecular Genetics, 14195 Berlin, Germany; 7Department of Medicine, University of Verona, 37129 Verona, Italy

**Keywords:** NF-kB, microRNAs, inflammation, data-mining

## Abstract

The nuclear factor NF-kB is the master transcription factor in the inflammatory process by modulating the expression of pro-inflammatory genes. However, an additional level of complexity is the ability to promote the transcriptional activation of post-transcriptional modulators of gene expression as non-coding RNA (i.e., miRNAs). While NF-kB’s role in inflammation-associated gene expression has been extensively investigated, the interplay between NF-kB and genes coding for miRNAs still deserves investigation. To identify miRNAs with potential NF-kB binding sites in their transcription start site, we predicted miRNA promoters by an in silico analysis using the PROmiRNA software, which allowed us to score the genomic region’s propensity to be miRNA cis-regulatory elements. A list of 722 human miRNAs was generated, of which 399 were expressed in at least one tissue involved in the inflammatory processes. The selection of “high-confidence” hairpins in miRbase identified 68 mature miRNAs, most of them previously identified as inflammamiRs. The identification of targeted pathways/diseases highlighted their involvement in the most common age-related diseases. Overall, our results reinforce the hypothesis that persistent activation of NF-kB could unbalance the transcription of specific inflammamiRNAs. The identification of such miRNAs could be of diagnostic/prognostic/therapeutic relevance for the most common inflammatory-related and age-related diseases.

## 1. Introduction

The nuclear factor (NF)-kB is a transcription factor (TF) activated by an evolutionarily conserved inflammatory signaling, induced by a wide range of external and internal danger signals [1,2,3]. The complex modulation of this signaling can be envisaged considering the different activation strategies, well known as “canonical” and “non-canonical” NF-kB activation signaling (reviewed in [4,5]). A fine-tuning activation of NF-kB promotes the expression of pro-inflammatory genes and participates in the regulation of survival, activation, and differentiation of innate immune cells and T cells [6]. On the contrary, a persistent activation of NF-kB signaling was described in conditions of cellular senescence and organismal aging, as well as in patients affected by the most common age-related degenerative diseases (ARDs) [7,8,9]. Many efforts have been made to understand which pathways are regulated by NF-kB and how the NF-kB pathway itself is modulated [10,11]. It has become clear that not only TFs but also a series of epigenetic factors, including non-coding microRNAs (miRNAs), are involved in the regulation of almost all the human transcriptional programs, both as inhibitors of mRNAs translation and as enhancers of mRNAs transcription [12,13,14]. Increasing evidence confirmed that these epigenetic factors play key roles in the development and progression of the most common human ARDs [15,16].

Regarding the canonical pathway of miRNA processing, that regulates gene expression at the post-transcriptional level, a primary transcript called pri-miRNA is cleaved to a precursor miRNA hairpin structure (pre-miRNA) in the nucleus by the Drosha/Pasha complex and transported into the cytoplasm, where the pre-miRNA is further processed into a miRNA:miRNA* duplex [17]. After being separated, the mature miRNA is loaded into the Argonaute 2 (Ago 2) containing RNA-induced silencing complexes (RISCs) and drives it to regulate its target mRNAs [17].

On one hand, a few miRNAs targeting mRNAs belonging to NF-kB pathway have already been identified, highlighting the activation of feedback loops aimed to restrain the inflammatory process triggered by NF-kB. Notably, some miRNAs involved in these feedback circuits were identified as deregulated in ARDs [5,18,19,20,21]. 

On the other hand, the full elucidation of miRNA biogenesis would be of paramount importance to identify their regulators and the role they might play in complex regulatory networks. Even if computational models were extensively applied to disentangle the complex effects of non-coding RNA in human diseases [22], for a long time, the difficulty of experimentally detecting miRNA promoters has limited the ability to identify the NF-kB binding sites in DNA sequences coding for miRNAs. However, the annotation of miRNA promoters, using high-throughput genomic data, allowed us to partially overcome this difficulty [23]. As important transcriptional regulators, miRNAs can upregulate or downregulate many target genes involved in the NF-kB signaling pathway via negative or positive feedback loops, and are involved in several human diseases, too, including the recent COVID-19 pandemic [24,25]. Since it is conceivable that age-related NF-kB activation could induce an overexpression of NF-kB responsive miRNAs, the identification of such miRNAs, and their targeted mRNAs and pathways, could contribute to clarifying the complex mechanisms that modulate healthy or unhealthy aging trajectories.

In this work, we aimed to: (i) identify all human miRNAs potentially modulated by NF-kB, (ii) select and characterize those NF-kB-responsive miRNAs that are specifically expressed in healthy tissues involved in the modulation of the inflammatory processes (such as cells of the immune system, liver, blood, and bone marrow), (iii) discover their targeted mRNAs and relative pathways, and finally (iv) evaluate the involvement of such pathways in the development of human diseases, including ARDs.

## 2. Results

### 2.1. Putative NF-kB Responsive miRNAs 

To select NF-kB responsive miRNAs, we analyzed the PROmiRNA database [23], FANTOM4 Libraries [26], “High confidence hairpins” in miRbase [27], and “Human expression dataset” [28], following the data-mining process highlighted in the data flow diagram in Figure 1.

We analyzed primarily genome-wide PROmiRNA predictions, as well as TF-binding site predictions as reported in [23], to identify miRNAs with potential NF-kB binding sites in their promoter sequences. PROmiRNA is a miRNA promoter recognition method, based on a semi-supervised statistical model trained on multi-tissue deepCAGE FANTOM4 libraries and other sequence features. It is tailored to score the potential of CAGE-enriched genomic regions to be promoters of either intergenic or intragenic miRNAs, thereby modulating miRNA expression in a tissue-specific manner [23]. To identify the TFs that regulate specific miRNAs, for each predicted miRNA transcription start site (TSS), we retrieved the 1 kb centered on it and used the TRAP approach [29] to compute the affinity of TF binding sites for all predicted miRNA promoters using TF models stored in the JASPAR database [30].

NF-kB appears among the first 10 TFs with the highest affinity for the 1000 bp-long region surrounding the predicted TSSs for 722 miRNA hairpin precursors (Appendix A). 

Since tissues show specific miRNA expression patterns, we aimed to highlight the list of putative NF-kB-responsive miRNAs expressed in tissues strictly involved in the modulation of the inflammatory processes, including inflammaging. To achieve this goal, we focused our subsequent research on those miRNAs transcribed in human tissues such as “T cells”, “T cells 2”, “monocytic-cells”, “immune system cells”, “bone marrow”, “blood”, and “liver”. Only the libraries relative to healthy tissues have been taken into consideration. This approach retrieved 399 miRNA hairpin precursors showing “expression at the promoter level” in at least one of these tissues (Appendix A). In general, this is a good indication that the mature forms of these miRNAs are expressed in a specific tissue. However, each step from DNA–RNA transcription to mature miRNA expression can be modulated, thereby modifying or blocking the final expression. Moreover, FANTOM4 libraries are characterized by a certain level of “transcriptional noise”, so we should expect false positives in mature miRNA predictions [23]. Therefore, among these putative NF-kB responsive miRNAs, we selected the “high confidence” hairpins in miRbase [27], retrieving 73 pre-miRNAs (Appendix A). A growing body of evidence suggests that mature sequences derived from both arms of the hairpin might be biologically functional and even that the dominant mature sequence can be processed from opposite arms [31,32]. Following the approach of selecting only the “high confidence” miRNA hairpins and filtering the dataset for “Human Expression dataset” [28], 68 “high confidence” expressed miRNAs were identified. This pool of miRNAs, reported in Table 1, constitutes our final set of putative NF-kB responsive miRNAs expressed in healthy tissues linked to inflammatory processes.

### 2.2. Genomic Features of Putative NF-kB Responsive miRNAs

According to their genomic location, it is possible to distinguish two classes of miRNAs: “intergenic miRNAs” are those located in intergenic regions of the genome, whereas “intragenic miRNAs” are those embedded in introns or exons of annotated genes [23]. Among the latter, “intronic miRNAs” are those located inside the introns of other genes and can either be co-transcribed with their host gene [33] or have an independent promoter [34,35,36], whereas intergenic miRNAs can derive from a primary miRNA transcript (pri-miRNAs) located in independent gene units [23,37]. Parallelly, it is possible to distinguish different categories of miRNA promoters: “intergenic promoters” are promoters assigned to intergenic miRNAs; “intragenic promoters” are promoters assigned to intragenic miRNAs and include both “host gene promoters” and “intronic promoters”; finally, “hybrid promoters” are those promoters that fall into intergenic regions upstream of intragenic miRNAs and could not be assigned unambiguously to the miRNA [23].

As shown in Table 1, among the promoter locations of the 68 putative NF-kB responsive miRNAs, 19 are “intergenic”, 15 are “host gene”, and 28 “intronic”. Interestingly, miR-15a, miR-16, miR-103, miR-186, and miR-33b can be modulated by both “host gene” and “intronic” promoters, whereas miR-194 is regulated by both “host gene” and “intergenic promoters”. Growing evidence indicates that alternative promoters are a mechanism for creating diversity in miRNA transcriptional regulation, as ascertained for protein-coding genes [38].

Regarding the phylogenesis of the 68 putative NF-kB responsive miRNAs, we showed that 22 miRNAs are conserved up to the vertebrate lineage (v), 38 miRNAs are conserved up to the mammal lineage (m), miR-194 and miR-19b up to the mammal and vertebrate lineage, and, finally, only 6 miRNAs are conserved in the primate lineage (p).

### 2.3. Characterization of the Interplay Linking NF-kB, miRNAs, and Their Host Genes

To better characterize miRNAs that share the promoters of the host gene and to determine whether those host genes are also known to be regulated by NF-kB, multiple assessments were conducted. Firstly, we retrieved available information regarding the host genes and their intragenic miRNAs, as reported in Table 2, whereas expression correlation plots between miRNAs and their host gene are shown in Appendix A.

No experimental evidence was found regarding the host gene of hsa-mir-374a, hsa-mir-545, or hsa-mir-15a. All the others are intronic miRNAs of genes involved in various biological processes ranging from DNA replication to differentiation:NFYC (Nuclear transcription factor Y subunit gamma) is a component of the sequence-specific heterotrimeric TF (NF-Y) which specifically recognizes a 5′- CCAAT-3′ box motif found in the promoters of its target genes. NF-Y can function as both an activator and a repressor, depending on its interacting cofactors [39];ZRANB2 (Zinc finger Ran-binding domain-containing protein 2) is a splicing factor required for alternative splicing of TRA2B/SFRS10 transcripts. May interfere with constitutive 5′-splice site selection [40];IARS2 (Isoleucine-tRNA ligase, mitochondrial) is a nuclear gene encoding mitochondrial isoleucyl-tRNA synthetase on which depends the translation of mitochondrial-encoded proteins [41];SMC4 (Structural maintenance of chromosomes protein 4) is the central component of the condensin complex, a complex required for the conversion of interphase chromatin into mitotic-like condense chromosomes [42];MCM7 (DNA replication licensing factor MCM7) acts as a component of the MCM2-7 complex (MCM complex) which is the replicative helicase essential for “once per cell cycle” DNA replication initiation and elongation in eukaryotic cells. It is the core component of CDC45-MCM-GINS (CMG) helicase, the molecular machine that unwinds template DNA during replication, and around which the replisome is built [43,44,45,46,47,48];NR6A1 (Nuclear receptor subfamily 6 group A member 1) is an orphan nuclear receptor that binds to a response element containing the sequence 5′-TCAAGGTCA-3′. By similarity, it may be involved in the regulation of gene expression in germ cell development during gametogenesis. It is involved in regulating embryonic stem cell differentiation, reproduction, and neuronal differentiation [49];TENM4 (Teneurin-4) is involved in neural development, regulating the establishment of proper connectivity within the nervous system. It plays a role in the establishment of the anterior–posterior axis during gastrulation. Moreover, it regulates the differentiation and cellular process formation of oligodendrocytes and myelination of small-diameter axons in the central nervous system (CNS) [50];COPZ1 (Coatomer subunit zeta-1) is a cytosolic protein complex involved in intracellular trafficking, endosome maturation, lipid homeostasis, and autophagy [51,52]. It is associated with iron metabolism through the regulation of transferrin [53,54];DDIT3 (DNA damage-inducible transcript 3 protein) is a multifunctional TF in endoplasmic reticulum stress response. It plays an essential role in the response to a wide variety of cell stresses and induces cell cycle arrest and apoptosis [55,56,57];WWP2 (NEDD4-like E3 ubiquitin-protein ligase WWP2) plays an important role in protein ubiquitination and inhibits activation-induced T cell death by catalyzing EGR2 ubiquitination [58]. In human embryonic stem cells, WWP2 promotes the degradation of TF OCT4, which not only plays an essential role in maintaining the pluripotent and self-renewing state of embryonic stem cells but also acts as a cell fate determinant through a gene dosage effect [55];HOXB3 (Homeobox protein Hox-B3) is a sequence-specific TF that is part of a developmental regulatory system that provides cells with specific positional identities on the anterior–posterior axis. Therefore, it may regulate gene expression, morphogenesis, and differentiation [59];SREBF1 (Sterol regulatory element-binding protein 1) is a precursor of the TF form (Processed sterol regulatory element-binding protein 1), which is embedded in the endoplasmic reticulum membrane [60]. Its processed form is a key TF that regulates the expression of genes involved in cholesterol biosynthesis and lipid homeostasis [60,61,62];PANK2 (Pantothenate kinase 2) is the mitochondrial isoform that catalyzes the phosphorylation of pantothenate to generate 4′-phosphopantothenate in the first and rate-determining step of coenzyme A (CoA) synthesis [63,64,65,66]. It is required for angiogenic activity of the umbilical vein of endothelial cells (HUVEC) [67].

Notably, five genes out of thirteen are engaged in transcription regulation (NR6A1, DDIT3, HOXB3, SREBF1, and NFYC), and only three are considered housekeeping genes (NFYC, ZRANB2, and COPZ1).

Experimentally validated interactions shared among the three groups of molecules, namely (i) the 21 NF-kB responsive miRNAs sharing the host gene promoter, (ii) their host genes, and (iii) the three TF members (NFKB1, REL, and RELA) are depicted in Figure 2.

Important nodes can be identified on the basis of their node centrality measures, such as degree and betweenness. The degree of a node is the total number of connections to other nodes. High-degree nodes are considered important “hubs” in a network [70,71]. The betweenness measures the number of shortest paths going through a node, taking into consideration the global network structure. Nodes with higher betweenness are important “bottlenecks” in a network [70,71]. Nodes identified by NFKB1, REL, miR-16-5p, miR-103a-3p, and NR6A1 have high degree centrality values, whereas RELA, miR-10a-5p, and miR-30e-5p represent nodes that occur between two dense clusters and have a high betweenness centrality even if their degree centrality values are not high.

Therefore, we performed an explorative evaluation of known and potential protein–protein interactions among REL, RELA, NFKB1, and miRNA-host genes (Figure 3) by querying the STRING Database [72,73,74].

The STRING network shows that almost all host gene proteins have some degree of interaction. Experimental and biochemical data confirm the functional association of NFKB1, REL, and RELA. On the other hand, the higher confidence interaction values suggest a functional link between DDIT3, NFYC, MCM7, and SREBF1, as well as between IARS2, SMC4, and WWP2. Of note, experimental evidence in Figure 2 indicated that NFBK1, REL, RELA, DDIT3, NFYC, MCM7, SREBF1, and SMC4 are all targets of miR-16-5p, but miR-103a-3p, in turn, regulates IARS2, MCM7, and WWP2.

Finally, the significantly differentially expressed host genes in ARDs have been identified (Table 3). Worth a mention is the downregulation of DDIT3, SMC4, and TENM4 in replicative senescence of human fibroblasts; the upregulation of SMC4 and MCM7 after vitamin C treatment; the upregulation of HOXB3 and TENM4 in Alzheimer’s disease; and the deregulation of DDIT3 and SMC4 in COVID-19 disease. 

### 2.4. Pathways Targeted by the 68 Putative NF-kB Responsive miRNAs

By performing an Ingenuity Pathway Analysis (IPA) Target Filter Analysis, we identified mRNAs targeted by the putative NF-kB responsive miRNAs. A total of 18,095 mRNAs were retrieved, of which 9613 were experimentally observed or highly predicted. The significance was reported as *p*-value in Appendix A. The let-7a-5p was the miRNAs with the highest associated number of mRNA targets (2014 targets).

Then we performed a network analysis focusing on putative NF-kB responsive miRNAs targeting mRNAs coding for molecules belonging to the NF-kB pathways (Figure 4).

Interestingly, the NF-kB responsive miRNAs do not directly target genes coding for the NF-kB different subunits, but most of them are able to target genes coding for molecules belonging to NF-kB activation pathways, such as TLR and MYD88. This result is very interesting, considering that the modulation of NF-kB biological activity is related to its activation, rather than to the modulation of NF-kB subunits expression.

Further, to discover the main diseases and functions associated with the selected miRNAs dataset, we performed an IPA Core Analysis (Figure 5). The diseases and functions are shown by bar chart, sorted by their −log *p*-value (*p*-value from Fisher’s Exact test).

Cancers, immunological diseases, neurological diseases, and metabolic diseases, all well-recognized as inflammatory-based diseases, are among the diseases associated with the highest probability with NF-kB responsive miRNAs. Focusing on metabolic diseases, the most affected diseases are the non-insulin dependent diabetes mellitus (−log *p*-value 11.955), Alzheimer disease (−log *p*-value 9.532), and diabetes mellitus (−log *p*-value 7.680).

To better explain the association of identified NF-kB putative responsive miRNAs with these human diseases, we depicted miRNAs-diseases relationship in Figure 6. Figure 6A depicts NF-KB putative responsive miRNAs associated with metabolic diseases, whereas Figure 6B–D, show the association between identified NF-kB responsive miRNAs and cardiovascular diseases, neurological diseases, and cancer, respectively.

### 2.5. The 68 Putative NF-kB Responsive miRNAs and Previously Identified Inflammamirs 

To test whether the 68 putative NF-kB responsive miRNAs could have a biological value in the context of the previous evidence, we compared our results with those already present in the literature. Among these 68 miRNAs, 21 have been experimentally validated to be transcribed by NF-kB1: miR-16-2 [75], miR-10a [76], miR-140-3p, miR-140-5p [77], miR-148b [78], miR-15b [79], miR-186 [80], miR-146a, miR-155, miR-19b, miR-20a, miR-19a, miR-17, miR-221, miR-222, miR-18a, miR-92a, miR-101, miR-23a, miR-27a, and miR-30c [21].

In addition, we have chosen as a reference all available data on the miRNAs relevant to aging, inflammation, and immunity that can be referred as inflammamiRs [81]. A detailed comparison table has been provided in Appendix A. Figure 7A shows the “word cloud” with the 68 “high confidence” expressed miRNAs. The more features a specific miRNA holds (such as: the number of promoter types, the number of miRNA precursors, if it is expressed in more than one tissue, and, finally, if it is known to target NF-kB), the bigger and bolder it appears in the figure. Figure 7B depicts a Venn diagram modified from [81], displaying the miRNAs related to inflammation, immunity, and aging based on their circulating shuttles.

In the inner circles are grouped exosome-associated miRNAs, while, in the outer circles, the circulating miRNAs associated with Ago-2, HDL, or other microparticles are grouped. In this version, it is important to note that bold characters indicate miRNAs overlapping among the two groups. Most of the 68 high-confidence NF-kB responsive miRNAs (reported in panel A) were previously identified as circulating miRNAs associated with aging, immunological functions, and inflammation, i.e., inflammaging [81]. Only three miRNAs, such as miR-154, miR-377, and miR-885-5p, were not retrieved in previous analysis [81]. However, based on recent literature, all of them are related to NF-kB/inflammation pathways [82,83,84]. All of the 68 NF-kB responsive miRNAs are therefore included in the Venn diagram reported in panel B, highlighting that these miRNAs identified as tissues expressed miRNAs are also detectable in blood, and most of them were identified inside extracellular vesicles, i.e., exosomes (miRNAs depicted in inner circles Figure 7B).

### 2.6. mRNAs Targeted by the 68 Putative NF-kB Responsive miRNAs Belonging to Pathways Involved in Aging Process and/or Age-Related Diseases

By further analyzing the IPA Target Filter Analysis results, we finally identified the mRNAs, either experimentally validated or highly predicted, to be targeted by the 68 putative NF-kB responsive miRNAs, belonging to pathways related to aging or to the most common ARDs. Among the 9613 mRNAs predicted to be targeted by such NF-kB responsive miRNAs, 189 mRNAs targeted by 46 out of 68 miRNAs were associated to “cellular senescence pathway” (Appendix A). In addition, out of the 9613, 8599 mRNAs were related to diseases reported in Figure 6, such as metabolic diseases, cardiovascular diseases, neurological diseases, and cancer. All these conditions share an inflammatory etiopathogenesis and are prototypical ARD.

## 3. Discussion

NF-kB is an ubiquitously and evolutionarily conserved TF activated by a plethora of external and internal proinflammatory stimuli [85,86,87]. The crucial role as a mediator of the inflammatory responses, together with the finding that the activation or inhibition of NF-kB can induce or reverse, respectively, the main features of aged organisms, has brought NF-kB under consideration as a key TF that drives the biological aging process [88]. In this framework, the identification of genes modulated by NF-kB can be considered a cutting-edge issue [89,90,91].

NF-kB-responsive genes were extensively investigated, whereas NF-kB-responsive genes for non-coding RNAs were only recently highlighted.

Here, we demonstrated that, by applying a data-mining approach, it is possible to select the most reliable NF-kB responsive miRNAs. Most notably, the availability of data on TFs binding sites on human miRNAs sequences constituted a starting point and the foundation for studying all human miRNAs with potential NF-kB binding sites in their promoter regions.

Some years ago, a general hypothesis was advanced that the aging process and the development of the most common ARDs could be fostered by a low-grade, chronic, systemic inflammatory process named “inflammaging” [92]. Inflammaging, which is principally sustained by the activation of the innate immune cells, is paralleled by the increased burden of senescent cells acquiring a senescence-associated secretory phenotype (SASP), which turns senescent cells into proinflammatory cells [86,93,94,95,96]. In immune cells and tissues obtained from patients affected by the most common ARDs, NF-kB is commonly constitutively activated [97]. Of note, NF-kB activation should be an inducible, but transient, event in physiological conditions. However, despite the presence of multiple checks and balances that control NF-kB activation, in cellular and organismal aging, as well as in many ARDs, NF-kB activation becomes persistent [98,99].

In this study, using PROmiRNA software and a data-mining approach, we provide a list of 73 putative “high confidence” pre-miRNAs sequences corresponding to 68 NF-kB responsive mature miRNAs sequences. 

Likewise, we highlighted the presence of distinct types of promoters that can regulate NF-kB responsive miRNAs. 

A total of 33 miRNAs of the 68 high confidence expressed miRNAs identified have an “intronic” promoter, and 5 of these have both an “intronic” and “host gene” promoter, whereas only one miRNA (miR-194) shares both “intergenic” and “host-gene” promoters. Alternative promoters are a common mechanism to create diversity in the transcriptional regulation of miRNA [38].

It has been demonstrated that “intronic” promoters convey an additional degree of freedom over intragenic miRNA transcriptional regulation by virtue of some peculiar characteristics, thus allowing the modulation of miRNA expression levels in a tissue- and condition-specific manner [23]. Besides the other features, in this context, it is important to stress that:“Intronic” promoters can explain cases of poor correlation between host gene and miRNA expression, functioning as a real alternative promoter [23]. As shown in Appendix A, the expression levels of NF-kB-miRNAs modulated by both “host gene” and “intronic” promoters (i.e., miR-16, miR-103, miR-186, and miR-33b) or by both “host gene” and “intergenic promoters” (i.e., miR-194) are not correlated with the expression levels of their host gene, whereas most of the miRNAs that share the host gene promoters are characterized by directly (e.g., miR-15b) or inversely (e.g., miR-30c, miR-616, and miR-93) correlated transcription levels.“Intronic” promoters are expressed in a tissue-specific manner, but “host gene” promoters are considered primarily for housekeeping gene regulation [23]. Housekeeping genes are required for the maintenance of essential functions of any cell type, so they are expected to be constitutively expressed in all cells and at any development stage [100]. Among the NF-kB-miRNA host genes, COPZ1, NFYC, and ZRANB2 have been cataloged as housekeeping genes (Table 2).“Intronic” promoters are mainly triggered by tissue-specific master regulator TFs, instead of TFs of “host gene” promoters, which broadly overlap with those of protein coding genes and can be considered mainly for housekeeping (“intergenic” promoters are regulated by a combination of intronic-specific and host-gene specific TFs). This suggests a different evolutionary mechanism [23]. In this study, the expression levels of the three housekeeping host genes (COPZ1, ZRANB2, and NFYC) and their miRNAs (miR-148b-3p, miR-186-5p, and, lastly, miR-30c-5p and miR-30e-5p, respectively) are mainly inversely correlated or not showing clear correlation trends (Appendix A).“Intronic” miRNA promoters are less evolutionarily conserved than either “intergenic” or “host gene” promoters [23].Conversely, evolutionarily conserved miRNAs are more likely to be regulated by an “intronic” promoter [23].Moreover, those intragenic miRNAs that share the promoters of the host gene interact with their own host genes (miR-16-2::MSC4; miR-106b::MCM7, miR-181b-2::NR6A1, miR-708::TENM4, miR-148b::COPZ1, and miR-10a::HOXB3), but also with the other functionally related host genes, creating a complex regulatory mechanism (Figure 2). NFKB1, REL, miR-16-5p, miR-103a-3p, and NR6A1 are the most important hub nodes in the network, whereas miR-10a-5p connects the hub nodes identified by NFKB1, NR6A1, and HOXB3, and miR-30e-5p connects REL, NR6A1, and ZRAMB2 hubs. Interestingly, in the network, it is possible to identify a clear TF-miRNA feed-forward loop involving DDIT3, miR-16-5p, and NFYC. In a TF-miRNA feed-forward loop, TF and miRNA co-regulate the target genes: in a “coherent” feed-forward loop, the TF and miRNA have the same effects on their common targets, whereas, in an “incoherent” feed-forward loop, the TF and miRNA carry out opposing effects, which precisely fine-tune gene expressions to minimize noise and maintain stability [68,101]. TF-miRNA feed-forward loops have a specific function in noise buffering effects, which can minimize the response of stochastic signaling noise to maintain steady-state target levels [102,103]. Disruption of feed-forward loops could lead to serious dysregulations at the origin of diseases and cancers, e.g., interference in the NF-kB/miR-19/CYLD loop can induce T cell leukemogenesis [103,104]. Therefore, investigating the regulatory motifs among DDIT3, 16-5p, and NFYC could provide valuable insights to dissect the molecular mechanisms underlying biological processes and diseases triggered by NF-kB constitutive activation.

Protein–protein interaction analysis of protein-coding host genes revealed that most of them could be functionally related (Figure 3). Beyond the well-known functional association of NFKB1, REL (cREL), and RELA, some data have highlighted the association with endoplasmic reticulum stress, providing opportunities to fine-tune cellular stress responses [105]. In the framework of atherosclerosis, multiple links between NF-kB and ER stress were suggested. A disturbed flow can cause endoplasmic reticulum stress, leading to SREBF1 activation with nuclear localization and to DDIT3 expression triggered by endoplasmic reticulum stress response elements [106]. NFYC is a subunit of a trimeric complex (NFY) known to interact with several TFs to enable the synergistic activation of specific classes of promoters. NFY directly controls the expression of TF genes such as P53 (DNA-damage), XBP1, CHOP/DDIT3 (endoplasmic reticulum stress), and HSF1 (heat shock) [107,108]. Of note, experimental data have shown the upregulation of both SMC4 and MCM7 in mesenchymal stem cells after vitamin C treatment; the downregulation of DDIT3, SMC4, and TENM4 in replicative senescence of human fibroblasts; the upregulation of HOXB3 and TENM4 in Alzheimer’s disease; and, finally, the deregulation of DDIT3 and SMC4 in COVID-19 disease (Table 3).

In this scenario, targeting NF-kB signaling is becoming a promising strategy for drug development and ARD treatments [91,109].

Almost all of the 68 miRNAs that we identified in our current analysis were previously associated with inflammaging processes and with the most common ARDs, such as metabolic diseases, cardiovascular diseases, neurodegenerative diseases, and cancers [110,111]. 

Out of the 9613 mRNAs targeted by the 68 NF-kB responsive miRNAs, 8599 mRNAs were related to such diseases. Of note, 189 mRNAs were associated with “cellular senescence pathway”, which is recognized as the main culprit of the aging process. 

Most of the NF-kB responsive miRNAs are involved in a negative feedback loop to restrain exacerbated inflammation [5,53,65,66,67,68,69,70].

Notably, the identified NF-kB responsive miRNAs are not able to directly modulate gene expression of NF-kB subunits but are able to target molecules belonging to NF-kB activation pathways (canonical and non-canonical pathway). Interestingly, among the NF-kB-responsive miRNAs genes identified with our approach, the most relevant examples of mRNAs that can target molecules belonging to the NF-kB canonical and non-canonical pathways, or related molecules, are miR-146a and miR-155. In fact, miR-146a and miR-155, control NF-kB activity during inflammation by a combinatory action without directly targeting NF-KB subunits [112]. miR-155 is rapidly upregulated by NF-kB during the early phase inflammatory response through a positive feedback loop necessary for signal amplification. miR-146a is rather gradually upregulated by NF-kB and forms a negative feedback loop attenuating NF-kB activity in the late phase of inflammation. The combined action of these two positive (NF-kB::miR-155) and negative (NF-kB::miR-146a) NF-kB-miRNA regulatory loops provides optimal NF-kB activity during inflammatory stimuli, and eventually lead to the resolution of the inflammatory response in physiological condition.

Another example is miR-16 that targets the IKKα/β complex of the NF-kB canonical pathway polarizing macrophages toward an M2 phenotype [113].These results are in line with the known modulation of NF-kB biological activity, based on the activation and not on the expression of its subunits [5].

Interestingly, all the 68 NF-kB responsive miRNAs are detectable in blood, and most of them were identified inside extracellular vesicles, i.e., exosomes. Exosomes are currently considered to be a crucial intercellular cross-talk mechanism, acting at both the paracrine and systemic levels [114]. This result highlights the complexity of the feed-back loops between NF-kB activation in specific tissues, the expression of NF-kB responsive miRNAs, and their release in the bloodstream as a systemic intercellular communication mechanism. A further level of complexity can be envisaged considering that NF-kB is known to indirectly regulate miRNA expression through the modulation of other TFs. NF-kB can modulate AP-1 TF [115], which, in turn, is able to modulate different miRNAs genes, i.e., miR-21 [116].

Of note, among the 68 miRNAs, 21 were already experimentally identified as NF-kB responsive, reinforcing the reliability of our results. Our data also highlight the potential value of the 47 NF-kB putative responsive miRNAs (listed in Figure 7B) that are yet to be experimentally validated.

Overall, our results are of interest in the framework of the research on the biomarkers/drugs of aging and inflammation related diseases. If NF-kB responsive miRNAs are hyper-transcribed in tissues involved in the modulation of inflammatory responses, the hypothesis that circulating miRNAs could be useful tools to track the trajectories of healthy or un-healthy aging is reinforced [117,118,119,120,121] and possible therapeutic strategies based on the inhibition of those miRNAs could be further tested.

### Limitation of the Study

The data-mining process frequently encompasses a further phase involving the extraction of implicit relational patterns through traditional statistics or machine learning, but the particularity of the research question and the type of data available have been a hindrance to this kind of analysis.

## 4. Methods

### 4.1. Data Mining Process

In the field of Knowledge Discovery in Databases (KDD), a data-mining approach is used to extract meaningful information and to develop significant relationships among variables stored in large data sets [122]. In this study, we have mined and integrated data from multiple databases to select NF-kB responsive miRNAs, and the process has been tailored based on the research question. Four main steps can be distinguished:

#### 4.1.1. Database Selection

The following data sources have been investigated to retrieve the data and develop the study: PROmiRNA [23], FANTOM4 libraries [26], “High confidence human hairpins” in miRBase [27], and “Human Expression” dataset (microrna.org) [28].

PROmiRNA provides an interesting approach for miRNA promoter annotation based on a semi-supervised statistical model trained on deepCAGE data and sequence features [23]. It was used to identify all human miRNAs potentially modulated by NF-kB, i.e., “NF-kappaB”, “NFKB1”, “REL”, and “RELA”.FANTOM4 libraries, generated by the FANTOM4 project [26], collects a wide range of genome-scale data from several tissues. The analysis of FANTOM4 libraries retrieved those miRNAs showing “expression at the promoter level” in different human tissues. The following libraries from healthy tissues were selected: “blood”, “bone marrow”, “immune system cells”, “liver”, “monocytic-cells”, “T cells”, and “T cells 2”.miRBase database is the public repository for all published miRNA sequences and associated annotations [27,32,123,124,125,126]. The “High confidence human hairpins” dataset [27] was downloaded to identify all human miRBase entries with high confidence levels assessed using the deep sequencing data sets collated in miRbase (The original dataset is provided in Appendix A, and it has been downloaded from this link: https://www.mirbase.org/blog/2014/07/high-confidence-mirna-set-available-for-mirbase-21/, accessed on 10 January 2023).Finally, microRNA.org [28], a comprehensive resource of miRNA target predictions and expression profiles, was searched to extract the “Human Expression dataset”, meaning the mature miRNA expression profiles in various tissues as presented by Landgraf et al. [127]. Expressed miRNAs from the following healthy tissues were selected (library names for each sample type are indicated in brackets): liver (hsa_Liver), pluripotent hematopoietic stem cell (hsa_HSC-CD34), B cells from peripheral blood (hsa_B-cell-CD19, hsa_B-cell-CD19-2, hsa_B-cell-CD19-pool), T-lymphocytes (hsa_T-cell-CD4, hsa_T-cell-CD4-2, hsa_T-cell-CD4-effector, hsa_T-cell-CD4-memory, hsa_T-cell-CD4-naïve, hsa_T-cell-CD8, hsa_T-cell-CD8-2→hsa_T-cell-CD8-naïve), NK cells (hsa_NK-CD56), monocytes (hsa_Monocytes-CD14), granulocytes (hsa_Granulocytes-CD15), and Dendritic cells (hsa_DC-unstim). Libraries from cell lines, tumor samples, genetic disorders, and so on, have been discharged. (The original dataset is provided in Appendix A, and it has been downloaded from this link: http://www.microrna.org/microrna/getDownloads.do, accessed on 10 January 2023).

#### 4.1.2. Data Extraction and Integration

This phase includes downloading, extracting, filtering, and combining the data from the databases previously identified. The integration of multiple datasets has been possible through the following steps.

#### 4.1.3. Data Cleaning and Transformation

Because the data originates from multiple sources, the integration often involves converting data formats, cleaning, removal of incorrect data, generating new variables, resolving redundancy, and checking against miRNA nomenclature consistency, both between miRNAs names originating in different miRBase versions and between the names of pri-miRNAs and the mature forms. This issue has been manually curated by comparing miRNA names in miRBase database version 21.

#### 4.1.4. Assessment of the Results

This is the final stage of a KDD process, involving the translation of aggregated data into comprehensible knowledge. The validity and reliability of the data were tested by comparing the results obtained in the data-mining process with those already published in the literature.

The whole data-mining process is illustrated in the data flow diagram in Figure 1. Data obtained at each intermediate step are provided in Appendix A. The final miRNA-pool is reported in Table 1.

### 4.2. Bioinformatic Evaluations

#### 4.2.1. Evaluation of miRNA-Host Gene-Transcription Factor Interactions

Host gene and intragenic miRNAs information (Table 2), as well as expression correlation data between miRNAs and their host genes (Appendix A), were retrieved from MiRIAD, a database integrating miRNA inter- and intragenic data (https://www.miriad-database.org/, accessed on 10 January 2023) [128]. In Table 2, host gene biological process were obtained from the UniProt database (Release 2022_05) (https://www.uniprot.org/, accessed on 10 January 2023) [129]; the Housekeeping and Reference Transcript Atlas (HRT Atlas v1.0) (https://housekeeping.unicamp.br/, accessed on 10 January 2023) [100] was investigated to discover those host genes cataloged as housekeeping genes.

Experimentally validated interactions shared among NFKB1, REL, RELA, the NF-kB-responsive miRNAs sharing the host gene promoter, and their host genes, were identified (Figure 2) by querying: (i) DIANA-TarBase v8 (http://www.microrna.gr/tarbase, accessed on 10 January 2023), retrieving experimentally supported miRNA-gene interactions [130]; (ii) TRRUST v2 (www.grnpedia.org/trrust, accessed on 10 January 2023), a manually curated database of transcriptional regulatory interactions [131]; and (iii) STRING v10, to highlight the protein–protein interactions, with the constraint to retrieve only experimental evidences [132]. The whole process, including the final network creation and visualization, was handled using miRNet (version 2.0), a miRNA-centric network visual analytics platform (https://www.mirnet.ca/, accessed on 10 January 2023) [68,70,133].

STRING database Version 11.5 (https://string-db.org/, accessed on 10 January 2023) was used to discover known and potential interactions among REL, RELA, NFKB1, and miRNA-host gene proteins (Figure 3). STRING is a database of predicted and known protein–protein interactions. The interactions include direct (physical) and indirect (functional) associations; these stem from knowledge transfer between organisms, from interactions aggregated from other (primary) databases, and from computational prediction [72,73,74]. The network was created by setting a minimum required interaction score of 0.15.

The RNA-seq datasets in Aging Atlas (https://ngdc.cncb.ac.cn/aging/index, accessed on 10 January 2023) were examined to explore age-related changes in host gene expression [134]. Table 3 shows differentially expressed host genes in strictly age-related conditions; only those genes showing |log2FC| > 1 and *q*-value < 0.005 (or *p*-value < 0.005 if *q*-value was not provided) have been reported. Data relative to particular experimental conditions (e.g., gene knockdown) have not been reported. All websites and online tools were accessed in the period between January and February 2023.

#### 4.2.2. Ingenuity Pathway Analysis

Bioinformatic evaluations (networks and disease analyses) were performed by the Ingenuity Pathway Analysis software (Qiagen, Hilden, Germany). The putative NF-kB responsive miRNAs identified through the data-mining process were analyzed to explore the experimentally observed or high predicted mRNA targets via the microRNA Target Filter Analysis.

Furthermore, an IPA Core Analysis was performed to define the associated diseases and functions. Direct and indirect relationships from the Ingenuity Knowledge Base (gene only) datasets were considered. We filtered only molecules and/or relationships experimentally observed in any tissue from human, rat, or mouse. Across the observations, 51 miRNAs were ready to be analyzed (Appendix A) [135]. All the networks, diseases, and biological functions were assessed using IPA software (Qiagen, Hilden, Germany).

## 5. Conclusions

Here, we demonstrated that a well-settled data-mining approach may disclose the most reliable miRNAs having a key role in the regulation of specific pathways of interest. Deciphering the crosstalk between miRNAs and NF-kB is one of the major topics to be investigated to understand the complex derailment of several metabolic pathways in normal and pathological aging. Future studies are needed to confirm that the identification of such miRNAs is of diagnostic/prognostic/therapeutic relevance for the most common inflammatory- and age-related conditions.

## Figures and Tables

**Figure 1 ijms-24-05123-f001:**
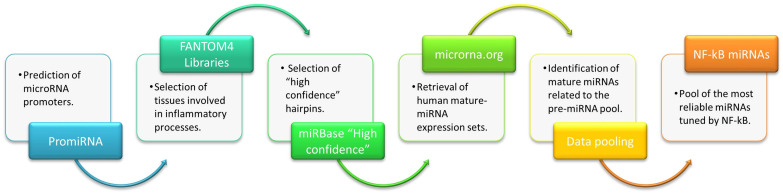
Data flow diagram. Figure depicts the whole data-mining process.

**Figure 2 ijms-24-05123-f002:**
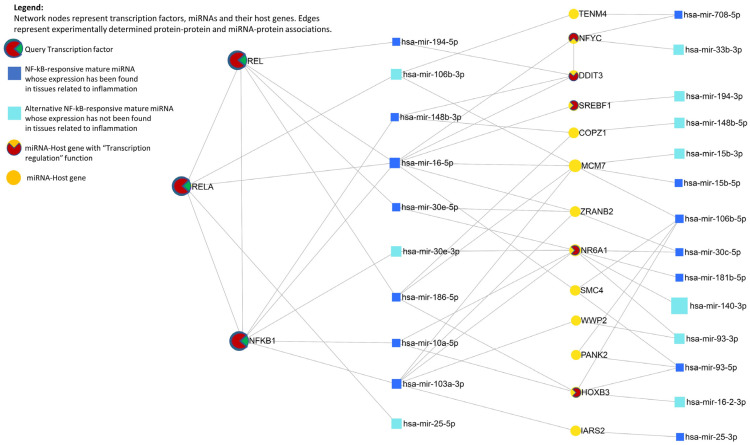
TF–miRNA co-regulatory network from experimentally validated data. In this visualization, a tripartite layout has been chosen. This provides an easy abstraction of relations between different types of molecular entities in complex networks composed of several types of nodes, such as miRNAs, genes, and TFs [68,69].

**Figure 3 ijms-24-05123-f003:**
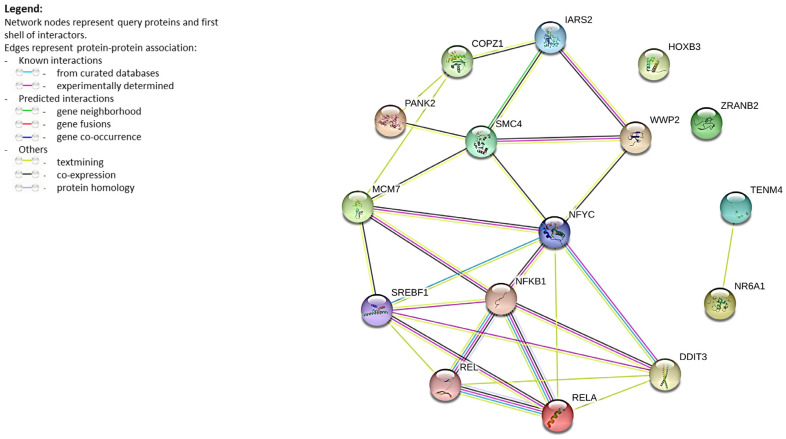
Protein–protein interaction network. Network nodes represent proteins: splice isoforms or post-translational modifications are collapsed, i.e., each node represents all the proteins produced by a single, protein-coding gene locus. Edges represent protein–protein associations and are meant to be specific and meaningful, i.e., proteins jointly contribute to a shared function; this does not necessarily mean they are physically binding to each other [72,73,74]. The greater the number of edges shared between two nodes, the greater the confidence of the interaction score. The line color indicates the type of interaction evidence.

**Figure 4 ijms-24-05123-f004:**
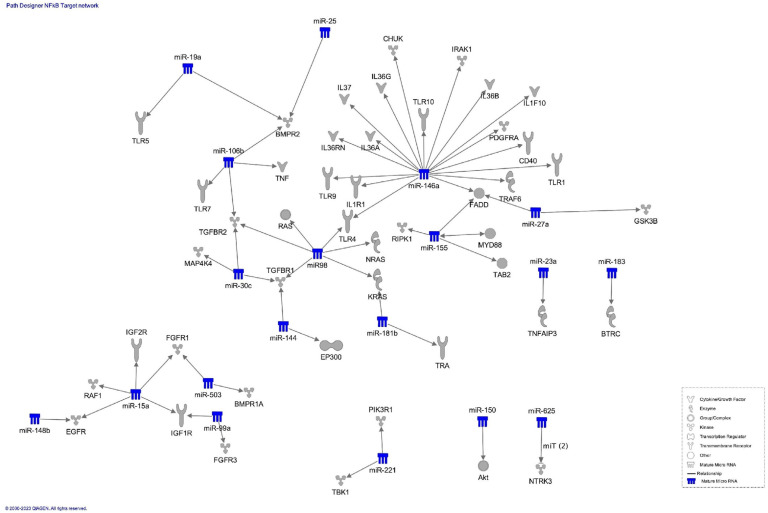
IPA Path Designer NF-kB target network. Molecules belonging to NF-kB pathway targeted by NF-kB responsive miRNAs. © 2000–2023 QIAGEN.

**Figure 5 ijms-24-05123-f005:**
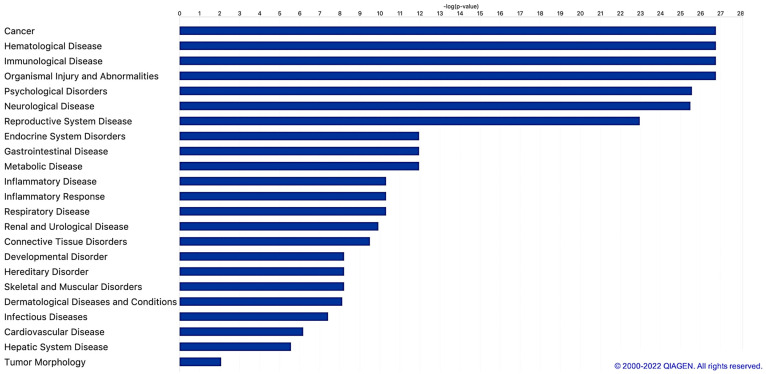
Most relevant human diseases associated with molecular pathways targeted by putative NF-kB responsive miRNAs. The diseases and functions associated with molecular pathways targeted by putative NF-kB responsive miRNAs are shown by the bar chart, sorted by their −log *p*-value (Fisher’s Exact test *p*-value). A total of 23 relevant human diseases are listed. © 2000–2023 QIAGEN.

**Figure 6 ijms-24-05123-f006:**
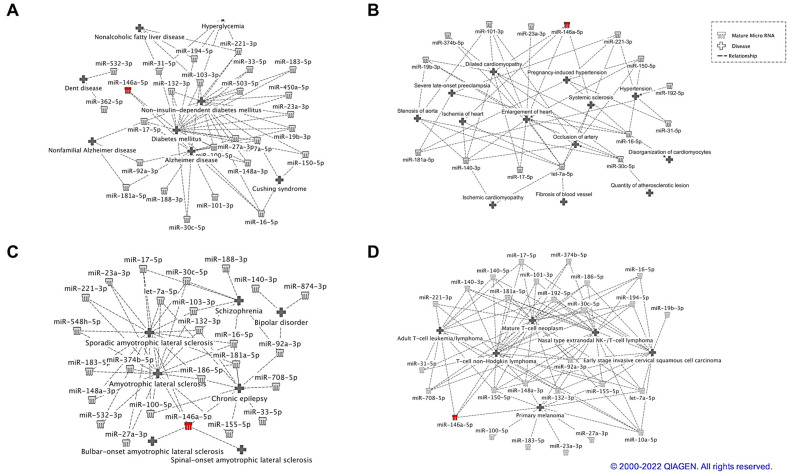
IPA diseases networks. Diseases networks targeted by NF-kB responsive miRNAs. Metabolic disease (panel (**A**)), cardiovascular diseases (panel (**B**)), neurological diseases (panel (**C**)), and cancer (panel (**D**)) © 2000–2023 QIAGEN.

**Figure 7 ijms-24-05123-f007:**
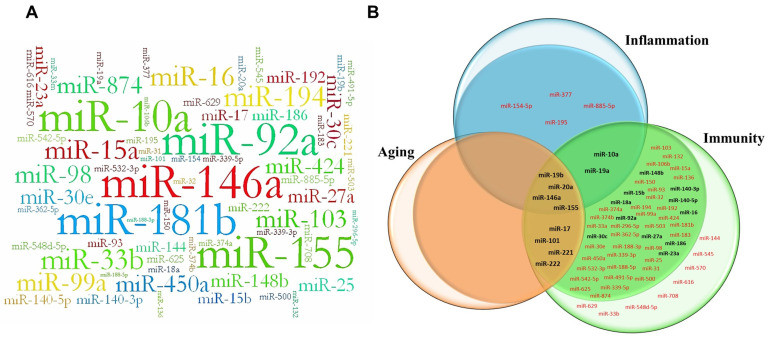
(**A**) Word cloud of the 68 putative NF-kB responsive miRNAs. The “word cloud” has been used to highlight the values of a miRNA based on its characteristics (such as: the number of promoter types, the number of miRNA precursors, if it is expressed in more than one tissue, and, finally, if it is known to target NF-kB). The more features a specific miRNA holds, the bigger and bolder it appears in the “word cloud”. This word cloud has been drawn using Wordaizer version 6.0 APP Helmond (www.apphelmond.com, accessed on 10 January 2023). (**B**) Venn diagram showing the NF-kB and inflammamiRs research in context. Modified version of the Venn diagram from [81]. The Venn diagram displays the 68 NF-kB responsive miRNAs related to inflammation, immunity, and aging based on their circulating shuttles. In bold, the 21 experimentally validated miRNAs; in red, the 47 not yet experimentally validated miRNAs. In the inner circles are grouped exosome-associated miRNAs, while in the outer circles are the circulating miRNAs associated with Ago-2, HDL, or other microparticles.

**Table 1 ijms-24-05123-t001:** The sixty-eight putative NF-kB responsive miRNAs expressed in healthy human tissues linked to inflammatory processes. The following attributes are reported: name of the mature miRNAs which derives from the pre-miRNAs previously identified, type of predicted TSS (“intergenic”, “host gene”, “intronic”, or “hybrid”), names of the healthy “Human Expression dataset” libraries in which the miRNAs are expressed (i.e., “liver” and “immune system”), the chromosome where the miRNA precursor is located, the age of the miRNAs corresponding to the predicted TSSs.

Putative NF-kB Responsive miRNAs *	Mature miRNA Expression ** in Tissues Linked to Inflammatory Processes	Prediction of Promoter Location According to PROmiRNA	miRNA Age	Chromosomes
hsa-miR-101	Immune system	intergenic	v	1
**hsa-miR-103**	Liver and Immune system	host gene-intronic	v	20
hsa-miR-106b	Immune system	host gene	m	7
hsa-miR-10a	Immune system	host gene	v	17
hsa-miR-132	Immune system	intergenic	m	17
hsa-miR-136	Immune system	intergenic	m	14
hsa-miR-140-3p	Liver and Immune system	host gene	v	16
hsa-miR-140-5p	Liver and Immune system	host gene	v	16
hsa-miR-144	Liver and Immune system	intergenic	v	17
hsa-miR-146a	Immune system	intronic	m	5
hsa-miR-148b	Immune system	host gene	m	12
hsa-miR-150	Immune system	intergenic	m	19
hsa-miR-154	Liver	intergenic	m	14
hsa-miR-155	Immune system	intronic	v	21
**hsa-miR-15a**	Liver and Immune system	host gene-intronic	v	13
hsa-miR-15b	Liver and Immune system	host gene	v	3
**hsa-miR-16**	Liver and Immune system	host gene-intronic	v	13|3
hsa-miR-17	Liver and Immune system	intronic	v	13
hsa-miR-181b	Immune system	host gene	v	9
hsa-miR-183	Immune system	intergenic	m	7
**hsa-miR-186**	Liver and Immune system	host gene-intronic	m	1
hsa-miR-188-3p	Immune system	intronic	m	X
hsa-miR-188-5p	Immune system	intronic	m	X
hsa-miR-18a	Immune system	intronic	v	13
hsa-miR-192	Liver and Immune system	intergenic	m	11
**hsa-miR-194**	Liver and Immune system	host gene-intergenic	m|v	11
hsa-miR-195	Liver	intronic	m	17
hsa-miR-19a	Immune system	intronic	v	13
hsa-miR-19b	Immune system	intronic	m|v	13
hsa-miR-20a	Immune system	intronic	v	13
hsa-miR-221	Immune system	intergenic	v	X
hsa-miR-222	Immune system	intergenic	v	X
hsa-miR-23a	Liver and Immune system	intergenic	m	19
hsa-miR-25	Liver and Immune system	host gene	m	7
hsa-miR-27a	Liver and Immune system	intergenic	m	19
hsa-miR-296-5p	Immune system	intergenic	m	20
hsa-miR-30c	Liver and Immune system	host gene	v	1
hsa-miR-30e	Liver and Immune system	host gene	v	1
hsa-miR-31	Immune system	intronic	v	9
hsa-miR-32	Immune system	intronic	v	9
hsa-miR-339-3p	Immune system	intronic	m	7
hsa-miR-339-5p	Immune system	intronic	m	7
hsa-miR-33a	Immune system	intronic	m	22
**hsa-miR-33b**	Immune system	host gene-intronic	m	17
hsa-miR-362-5p	Immune system	intronic	m	X
hsa-miR-374a	Immune system	host gene	m	X
hsa-miR-374b	Immune system	intergenic	m	X
hsa-miR-377	Liver	intergenic	m	14
hsa-miR-424	Liver and Immune system	intergenic	m	X
hsa-miR-450a	Immune system	intergenic	m	X
hsa-miR-491-5p	Immune system	intronic	m	9
hsa-miR-500	Liver	intronic	m	X
hsa-miR-503	Immune system	intergenic	m	X
hsa-miR-532-3p	Immune system	intronic	m	X
hsa-miR-542-5p	Immune system	intergenic	m	X
hsa-miR-545	Immune system	host gene	m	X
hsa-miR-548d-5p	Immune system	intronic	p	8
hsa-miR-570	Immune system	intronic	p	3
hsa-miR-616	Immune system	host gene	p	12
hsa-miR-625	Immune system	intronic	p	14
hsa-miR-629	Immune system	intronic	p	15
hsa-miR-708	Immune system	host gene	m	11
hsa-miR-874	Liver and Immune system	intronic	m	5
hsa-miR-885-5p	Liver	intronic	p	3
hsa-miR-92a	Liver and Immune system	intronic	m	13
hsa-miR-93	Immune system	host gene	m	7
hsa-miR-98	Liver and Immune system	intronic	m	X
hsa-miR-99a	Liver and Immune system	intronic	v	21

Note: miRNAs highlighted in bold are those processed starting from two or more pre-miRNA hairpins, each one transcribed starting from two different promoter types. ***** In PROmiRNA, NF-kB is among the top 10 TFs with the highest affinity for the 1000 bp-long region surrounding the predicted TSSs. ****** Mature miRNAs have been selected based on the “Human Expression dataset” (microrna.org, accessed on 10 January 2023). This selection allows to review mature miRNA expression patterns across the tissues of interest.

**Table 2 ijms-24-05123-t002:** Host genes and intragenic miRNAs information.

miRNA Hairpin	Host Gene Information				Intragenic miRNA Information			
	Ch	Host Gene	Entrez Gene ID	Host Gene Biological Process	Intron n°	Distance from Upstream Exon	Direction	Mature miRNA ID in miRBase 22.1 Release (In Parentheses Previous IDs) §
hsa-mir-30c-1 ^H^	1	NFYC	4802	Transcription regulation ‡	4	4038	sense	miR-30c-5p (**miR-30c**); miR-30c-1-3p (miR-30c-1*)
hsa-mir-30e ^H^	1	NFYC	4802	Transcription regulation ‡	4	1109	sense	miR-30e-5p (**miR-30e**); miR-30e-3p (miR-30e*)
hsa-mir-186 ^H, I^	1	ZRANB2	9406	mRNA processing ‡	8	1560	sense	miR-186-5p (**miR-186**); miR-186-3p (miR-186*)
hsa-mir-194-1 ^H, Ig^	1, 11	IARS2	55699	Protein biosynthesis	12	6996	antisense	miR-194-5p (**miR-194**); miR-194-3p (miR-194*)
hsa-mir-15b ^H^	3	SMC4	10051	DNA condensation	3	84	sense	miR-15b-5p (**miR-15b**); miR-15b-3p (miR-15b*)
hsa-mir-16-2 ^H, I^	3, 13	SMC4	10051	DNA condensation	3	241	sense	miR-16-5p (**miR-16**); miR-16-2-3p (miR-16-2*)
hsa-mir-106b ^H^	7	MCM7	4176	DNA replication	13	99	sense	miR-106b-5p (**miR-106b**); miR-106b-3p (miR-106b*)
hsa-mir-25 ^H^	7	MCM7	4176	DNA replication	13	530	sense	miR-25-5p (miR-25*); miR-25-3p (**miR-25**)
hsa-mir-93 ^H^	7	MCM7	4176	DNA replication	13	326	sense	miR-93-5p (**miR-93**); miR-93-3p (miR-93*)
hsa-mir-181b-2 ^H^	9	NR6A1	2649	Transcription regulation	2	139,140	antisense	miR-181b-5p (**miR-181b**); miR-181b-2-3p
hsa-mir-708 ^H^	11	TENM4	26011	Differentiation	1	38,400	sense	miR-708-5p (**miR-708**); miR-708-3p (miR-708*)
hsa-mir-148b ^H^	12	COPZ1	22818	Protein transport ‡	1	12,035	sense	miR-148b-5p (miR-148b*); miR-148b-3p (**miR-148b**)
hsa-mir-616 ^H^	12	DDIT3	1649	Transcription regulation	1	1159	sense	miR-616-5p (miR-616, miR-616*); miR-616-3p (**miR-616**)
hsa-mir-140 ^H^	16	WWP2	11060	Ubl conjugation pathway	6	1191	sense	**miR-140-5p** (miR-140); **miR-140-3p**
hsa-mir-10a ^H^	17	HOXB3	3213	Transcription regulation	1	2202	sense	miR-10a-5p (**miR-10a**); miR-10a-3p (miR-10a*)
hsa-mir-33b ^H, I^	17	SREBF1	6720	Transcription regulation	12	314	sense	miR-33b-5p (**miR-33b**); miR-33b-3p (miR-33b*)
hsa-mir-103a-2 ^H, I^	20	PANK2	80025	Coenzyme A biosynthesis	5	444	sense	miR-103a-2-5p (miR-103-2*; miR-103a-2*); miR-103a-3p (**miR-103**, miR-103a)

Note: Promoter location according to PROmiRNA: **^H^** host gene; **^I^** intronic; **^Ig^** intergenic. **Ch**: chromosome number; in bold, the locus of interest if more than one is indicated. **‡** Genes classified as housekeeping gene in Housekeeping and Reference Transcript Atlas. **§** In the last column, mature miRNAs nomenclature has been harmonized throughout miRBase database; miRNA nomenclature used in Table 1 has been highlighted in bold.

**Table 3 ijms-24-05123-t003:** Differentially expressed miRNA-host genes in age-related diseases.

Differentially Expressed Gene	Cell/Tissue	Treatment/Condition	log2 Fold Change	*p*-Value	*q*-Value	DOI
DDIT3	Human diploid fibroblasts IMR90	Replicative senescence	−1.09474	1.55 × 10^−23^	2.63 × 10^−22^	10.1093/nar/gkz555
DDIT3	Lung	COVID-19 vs. Control	3.04617	1.11 × 10^−17^	2.55 × 10^−16^	10.1038/s41556-021-00796-6
HOXB3	Human induced pluripotent stem (iPS) cell-derived neural progenitor cells	Alzheimer’s disease	1.30000	5.10 × 10^−4^	3.40 × 10^−2^	10.1016/j.celrep.2019.01.023
MCM7	Human arterial endothelial cell	Ionizing radiation	−1.07367	2.66 × 10^−33^	1.94 × 10^−31^	10.1093/nar/gkz555
MCM7	Human diploid fibroblasts WI38	Ionizing radiation	−1.60665	1.53 × 10^−10^	5.99 × 10^−9^	10.1093/nar/gkz555
MCM7	WRN-/- mesenchymal stem cell	Vitamin C	1.90980	5.08 × 10^−51^		10.1007/s13238-016-0278-1
SMC4	Human arterial endothelial cell	Ionizing radiation	−1.14304	5.14 × 10^−38^	4.56 × 10^−36^	10.1093/nar/gkz555
SMC4	Human diploid fibroblasts WI38	Ionizing radiation	−2.31474	1.24 × 10^−14^	8.41 × 10^−13^	10.1093/nar/gkz555
SMC4	Lung	COVID19 vs. Control	−2.85324	4.44 × 10^−10^	3.79 × 10^−9^	10.1038/s41556-021-00796-6
SMC4	Human diploid fibroblasts WI38	Replicative senescence	−1.44078	8.14 × 10^−7^	1.75 × 10^−5^	10.1093/nar/gkz555
SMC4	WRN-/- mesenchymal stem cell	Vitamin C	2.18610	4.24 × 10^−55^		10.1007/s13238-016-0278-1
TENM4	Human diploid fibroblasts WI38	Replicative senescence	−4.84098		1.97 × 10^−52^	10.1093/nar/gkz555
TENM4	Human diploid fibroblasts IMR90	Ionizing radiation	−1.63819	3.26 × 10^−23^	5.65 × 10^−22^	10.1093/nar/gkz555
TENM4	Human diploid fibroblasts WI38	Ionizing radiation	1.82630	6.17 × 10^−12^	3.09 × 10^−10^	10.1093/nar/gkz555
TENM4	Human diploid fibroblasts IMR90	Replicative senescence	−1.03845	1.66 × 10^−10^	1.24 × 10^−9^	10.1093/nar/gkz555
TENM4	Human induced pluripotent stem (iPS) cell-derived neurons	Alzheimer’s disease	1.70000	5.50 × 10^−4^	1.70 × 10^−2^	10.1016/j.celrep.2019.01.023

## Data Availability

Not applicable.

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
