# Peer review of "A Data-Mining Approach to Identify NF-kB-Responsive microRNAs in Tissues Involved in Inflammatory Processes: Potential Relevance in Age-Related Diseases"

_ijms, 2023, doi:10.3390/ijms24065123_

Round 1

Reviewer 1 Report

The manuscript amdescribes a process to select 68 putative NF-kB responsive miRNAs. Then the authors identified associated pathways/diseases  by performing IPA analysis. For all the results are based on silico analysis, the relaible of predictions is very important. The following is my comments:

1. The authors evaluates the prediction reliable by comparing with those already published in literatures. Actually the predictions are compared with inflammamiRs,which were publised in 2017. That means the authors didn't  include the literatures published in the past five years.  

2. How many putative NF-kB responsive miRNAs are not in previously identified inflammamiRs? Are these miRNAs false positive?

3. There are some typos in manuscripts.Such as line 141 ' a DDK process ' should be ' a KDD process', line 297 'displaing' shuold be 'displaying'.

Author Response

We are grateful to the reviewers and the editor for their constructive comments and suggestions, which helped to improve the quality of this manuscript. We are going to thoroughly discuss the points raised by the reviewers. Changes in the manuscript are highlighted in red. We hope that the revised manuscript is suitable for publication in IJMS.

Point-by-point responses to the reviewer’s comments:

Reviewer 1

The manuscript describes a process to select 68 putative NF-kB responsive miRNAs. Then the authors identified associated pathways/diseases by performing IPA analysis. For all the results are based on silico analysis, the reliable of predictions is very important. The following is my comments:

  1. The authors evaluates the prediction reliable by comparing with those already published in literatures. Actually the predictions are compared with inflammamiRs,which were publised in 2017. That means the authors didn't include the literatures published in the past five years.  

We thank the reviewer for the assessment of our manuscript and for the suggestions. Our group provided evidence that some miRNAs involved in the modulation of inflammation are deregulated in aging process as well as in age-related diseases (ARDs). We tagged such miRNAs as inflammamiRs (Prattichizzo et al., 2017). Ven diagram published in 2017 that we reported modified in the submitted manuscript, included circulating miRNAs, either exosome-borne or protein-bound, able to modulate inflammation, immunity, and aging pathways. By reporting these data in the present manuscript, we want to highlight that most of the 68 putative NF-KB modulated miRNAs, selected in silico among those expressed in tissues related to inflammation, immune system and liver, were the same previously described as circulating microRNAs associated with the modulation of inflammation, immunity, and aging pathways. Only 3 miRNAs, miR-154, -377, and -885-5p, were not an output of our previous analysis (46). We searched for additional information about these 3 miRNAs and high-lighted that all of them are related to NF-kB/inflammation pathways. All the 68 NF-kB responsive miRNAs are therefore included in Venn diagram reported in panel B, concluding that these miRNAs identified as tissues expressed miRNAs in the present study, are also detectable in blood and most of them were identified inside extracellular vesicles, i.e., exosomes (miRNAs depicted in inner circles Fig. 7 panel B. In addition, to test whether the putative 68 NF-kB responsive miRNAs could have a biological value in the context of the previous evidence, we compared our results with those in the literature. Among the 68 miRNAs, 21 have been experimentally validated to be transcribed by NF-kB1. The paragraph 2.5. entitled “68 putative NF-kB responsive miRNAs and previously identified inflammamiRs” was extensively revised, accordingly.

  1. How many putative NF-kB responsive miRNAs are not in previously identified inflammamiRs? Are these miRNAs false positive?

As reported above, all the 68 miRNAs that we identified as NF-kB responsive miRNAs, were previously identified as inflammamiRs. We upgraded Figure 7 panel B of revised manuscript. 

  1. There are some typos in manuscripts.Such as line 141 ' a DDK process ' should be ' a KDD process', line 297 'displaing' shuold be 'displaying'.

The manuscript was revised for typos and grammar and improved. As pointed out by the two reviewers, in line 141 a "DDK process" was changed to "a KDD process"; in line 297 "displaing" was changed to "displaying"; in line 288 "contest" was changed to "context".

Author Response

We are grateful to the reviewers and the editor for their constructive comments and suggestions, which helped to improve the quality of this manuscript. We are going to thoroughly discuss the points raised by the reviewers. Changes in the manuscript are highlighted in red. We hope that the revised manuscript is suitable for publication in IJMS.

Point-by-point responses to the reviewer’s comments:

Reviewer 2

The manuscript “A data-mining approach to identify NF-kB-responsive microRNAs in tissues involved in inflammatory processes: potential relevance in age-related diseases” by Micolucci et al. identifies NF-kB-regulated microRNAs (miRNAs) that are expressed in healthy human tissues, especially those tissues that are associated with inflammation. The authors also described how the potential targets of these miRNAs are involved in biological processes and pathways associated with age-related conditions and diseases.

Overall I enjoyed reading the manuscript and have very specific comments.

We thank the reviewer for the assessment of our manuscript and for the interesting suggestions.

  1. For miRNAs that share the host gene promoters, are those host genes also known to be regulated by NF-kB?

A new paragraph, entitled “2.3. Characterization of the interplay linking NF-kB, miRNAs, and their host genes” and new figures and tables were added in the revised manuscript to better address this issue.

As suggested by the reviewer, to better characterize miRNAs that share the promoters of the host gene and to determine whether those host genes are also known to be regulated by NF-kB, we conducted multiple assessments. Firstly, we retrieved available information regarding the host genes and their intragenic miRNAs as reported in new Table 2 whereas, expression correlation plots between miRNAs and their host gene are shown in new Figure S1. In addition, experimentally validated interactions shared among the three group of molecules, namely i) the 21 NF-kB-responsive miRNAs sharing the host gene promoter, ii) their host genes, and iii) the three TF members (NFKB1, REL and RELA) are depicted in the new Figure 2. Protein-Protein Interaction Network is depicted in the new Figure 3. Finally, the new Table 3 reports the differentially expressed miRNA-host genes in age-related conditions.

All these new data were critically discussed in the discussion section of the revised manuscript.

  1. The major targets identified from this study (e.g., miR-146a, miR-155 etc.) are already known and validated by multiple other groups. So re-identifying them through data mining is of very little value (well I agree this proves the efficiency of the data mining approach). I think the authors should highlight the ones that are yet to be validated experimentally which will be a new addition to the field.

Among the 68 NF-kB-responsive microRNAs, 21 have been previously experimentally validated to be transcribed by NF-kB1. 47 microRNAs are yet to be validated experimentally. This setting of miRNAs was highlighted in red in figure 7 panel B of the revised manuscript.

3.Apart from directly regulating the miRNAs at the transcription level, NF-kB is known to indirectly regulate miRNA expression through other transcription factors. This should be mentioned in the discussion.

According to reviewer’s comment we discuss this issue in the revised discussion section. As an example of this further level of complexity we reported that NF-kB can modulate AP-1 transcription factor, which in turn is able to modulate different miRNAs genes, i.e. miR-21, that can modulate NF-KB pathway.

4.Out of the 9613 mRNAs that were either experimentally validated or highly predicted to be targeted by these NF-kB-responsive miRNAs, how many overlap with genes whose expression is decreased with aging or other age-related conditions?

As suggested by the reviewer we identified among the results of the IPA Target Filter Analysis   the mRNAs targeted by the 68 putative NF-kB responsive miRNAs belonging to pathways involved in senescence and/or age-related diseases. The results were reported in a new supplementary Table (supplementary Table 6) and described in a new paragraph entitled “3.5 mRNAs targeted by the 68 putative NF-kB responsive miRNAs belonging to pathways involved in aging process and/or age-related diseases”. In detail we stated that: “By further analyzing the IPA Target Filter Analysis results, we finally identified the mRNAs, either experimentally validated or highly predicted, to be targeted by the 68 putative NF-kB responsive miRNAs, belonging to pathways related to aging or to the most common age-related conditions. Among the 9613 mRNAs predicted to be targeted by such NF-kB responsive miRNAs, 189 mRNAs targeted by 46 out of 68 miRNAs, were associated to “cellular senescence pathway” (Table S6). In addition, out of the 9613, quite all, 8599 mRNAs were related to diseases reported in Figure 6, such as metabolic disease, cardiovascular diseases, neurological diseases, and cancer. All these conditions share an inflammatory etiopathogenesis and are prototypical age-related diseases.”

5.There are some typos and grammatical errors (e.g., line 288 would be ‘context’ not ‘contest’; line 297 would be ‘displaying’ not ‘displaing’ etc.) that need to be corrected.

The manuscript was revised for typos and grammar and improved. In particular, as pointed out by the two reviewers, in line 141 a "DDK process" was changed to "a KDD process"; in line 297 "displaing" was changed to "displaying"; in line 288 "contest" was changed to "context".
